# Comparative Evaluation of Esthetic and Structural Aspects in Class II Functional Therapy. A Case–Control Retrospective Study

**DOI:** 10.3390/ijerph18136978

**Published:** 2021-06-29

**Authors:** Gabriella Galluccio, Rosanna Guarnieri, Diana Jamshir, Alessandra Impellizzeri, Gaetano Ierardo, Ersilia Barbato

**Affiliations:** Department of Oral and Maxillofacial Sciences, Sapienza University of Rome, 00161 Rome, Italy; gabriella.galluccio@uniroma1.it (G.G.); dianajamshir@yahoo.it (D.J.); alessandra.impellizzeri@uniroma1.it (A.I.); gaetano.ierardo@uniroma1.it (G.I.); ersilia.barbato@uniroma1.it (E.B.)

**Keywords:** Class II malocclusion, functional therapy, Fränkel-2, preformed appliances, Twin Block, mandibular growth, esthetic analysis, Occlus-o-Guide^®^

## Abstract

**Background:** To compare the skeletal, dental, and esthetic changes produced by three functional devices, Fränkel-2 appliance (FR-2), Twin Block (TB), and Occlus-o-Guide^®^ (O-o-G^®^), for the treatment of Class II malocclusion. **Methods**: Sixty-five patients with Class II Division 1 malocclusion were divided into three groups and were analyzed through cephalometric analysis of skeletal, dental, and esthetic variables before and after treatment. The first group of 23 patients (F: 9; M: 14; mean age: 10.3 ± 1.08 years) was treated with FR-2, the second group of 18 patients (F: 8; M: 10; mean age 10.7 ± 1.05 years) was treated with TB, the third group (F: 11; M: 13; mean age: 9.05 ± 0.39 years) of 24 patients was treated with O-o-G^®^. The structural effects of the three devices were compared with a control group generated by the growth variations reported in the cephalometric atlas of Bhatia and Leighton. Esthetic analysis was performed comparing the results of the patients treated with a control group of 20 subjects with mandibular retrognathia and Class II Division 1 malocclusion, not subjected to therapy. **Results**: The three devices resulted in a significant increase in mandibular length, with higher results obtained for FR-2 and TB. A statistically significant increase in the IMPA angle was found for the O-o-G^®^ group, and a notable reduction of both overjet and overbite was detected in all three groups of treated patients. The esthetic evaluation showed overall more significant results in the TB group, especially with regard to the reduction of facial convexity. The retrusion of the upper lip was on average more significant in the O-o-G^®^ group, followed by that in the TB. **Conclusions**: All three devices have proven to be effective overall in resolving skeletal changes and improving facial esthetics.

## 1. Introduction

Class II malocclusions represent a condition frequently found in the population and in orthodontic practice [1]. Class II functional appliances are designed to position the jaw forward and down to modify its growth by the pressure exerted by soft tissues to bring the jaw back to its ideal position. Subjects suffering from Class II Division I malocclusion are generally characterized by facial convexity, increased overjet, labial incompetence, and often an unattractive facial profile that could induce the subject to not have a satisfactory self-image. Such esthetic defects often cause negative emotional impact in the growing child and could hinder the establishment of interpersonal relationships [2,3,4,5]. It is important to remember that among the main objectives of orthodontic treatment there should not only be the resolution of skeletal and dental discrepancies but also the improvement of facial esthetics [6,7].

Orthodontists use functional therapy in growing Class II patients in the orthopedic phase of orthodontic treatment. Costs, patient compliance, ease of management, and time to achieve a Class I are all factors that influence the specialist’s therapeutic choice. The three devices evaluated in our study are effective in determining mandibular growth, which is the main expected effect in the treatment of Class II, and in achieving occlusal outcome. Most patients tolerate these devices well, but the compliance is one of the factors that most affects the outcome of the treatment. The correct timing and correct management of a Class II functional therapy reduce the biological cost in terms of subsequent fixed therapy times and improve the overall final result [8].

Rolf Fränkel conceived his functional regulator exclusively for tissue retention around 1956, referring faithfully to the theory of the functional matrix of Moss. The Fränkel-2 consists of vestibular shields designed to keep the jaw in a protruded position thanks to neuromuscular reprogramming, configuring itself as an involuntarygymnastics instructor [9,10].

The Twin Block was presented by Clark around 1982 and consists of double plates, upper and lower, equipped with occlusal resin bites (bite blocks) in contact on inclined planes with an angle of about 70°, in the distal region of the second lower premolars, resulting in a lower and forward postural displacement of the jaw [11,12].

The Occlus-o-Guide^®^ is a pre-formed elastodontic functional device of interceptive orthodontics. It can be made in different series, of which the G series is the most popular; this design accommodates the presence of specific niches for each dental element in order to guide its correct eruption and ensure its alignment [13,14,15].

The aim of the study is to compare the skeletal, dental, and esthetic changes produced by three functional devices, Fränkel-2 appliance (FR-2), Twin Block (TB), and Occlus-o-Guide^®^ (O-o-G^®^), for the treatment of Class II malocclusion.

## 2. Materials and Methods

### 2.1. Subjects

The patients were screened and treated in the Department of Oral and Maxillofacial Sciences, Sapienza University of Rome (UOC of Orthodontics and UOC of Pediatric Dentistry) between 2018 and 2020 according to a standardized and existing protocol that provided for: an initial visit with orthopantomography and teleradiography; an orthodontic treatment with one of three functional appliances: Fränkel-2 (FR-2), Twin Block (TB), and Occlus-o-Guide (O-o-G^®^); and a follow-up visit 4–6 months after the end of their treatment.

From the original study sample of 1102 subjects with Class II Division 1 malocclusion, a group of 65 patients (28 females and 37 males, ages ranged from 8.3 to 12.8 years, mean: 9.95 years) was selected. The inclusion criteria were as follows: skeletal Class II Division 1 relationship (ANB ≥ 4°); overjet ≥3 mm; SN–GoGn = 32° ± 6°; minimal crowding in dental arches (≤4 mm); bilateral Class II molar and canine relation; growing patients (stages of cervical vertebrae from CS2 to CS3). The exclusion criteria were as follows: previous orthodontic treatment; severe maxillary transverse deficiency; severe skeletal asymmetry; lack of compliance; systemic diseases (metabolic disorders, syndromic pathologies, or other conditions that may affect the orthodontic treatment results).

The investigation was reviewed and approved by the regional Ethical Review Board of the Umberto I General Hospital (No. 3802).

At the time T1, the first group of 23 patients treated with FR-2 included 9 females and 14 males (mean age of 10.3 ± 1.08 years), the second group of 18 patients treated with TB included 8 females and 10 males (mean age of 10.7 ± 1.05 years), the third group of 24 patients treated with Occlus-o-Guide^®^ included 11 females and 13 males (mean age of 9.05 ± 0.39 years). The need to have a reference group as homogeneous as possible to the sample of treated patients led us to select two different control groups, respectively for the structural and esthetic evaluation.

The structural effects have been analyzed in relation to the expected changes from a standard growth pattern, comparing them with a control group generated by the growth variations reported in the Bhatia and Leighton cephalometric atlas [16]. The average age at T1 was 10 years and two months, the time elapsed between T1 and T2 was 12 months. The possibility of having a control group of untreated patients, with the same malocclusion of the subject’s undergoing therapy, is limited by ethical considerations [17].

The esthetic analysis of the patients treated with one of the three functional devices was carried out by comparing the results obtained with those of a control group of 20 patients affected by mandibular retrognathia Class II Division I malocclusion, not treated with any therapy analyzed in the study of Baysal and Uysal [6]. The control group included 9 females and 11 males (mean age of 12.17 ± 1.7 years) and the period between T1 and T2 time was 15.58 ± 3.13 months. Twenty patients of the control group were statistically compared with the three case groups separately (23 patients for FR-2 group; 18 patients for TB group, and 24 patients for O-o-G^®^).

### 2.2. Cephalometric Analysis

The cephalometric analysis was performed on lateral cephalometric radiographs at times T1 and at T2. For each subject included in the sample, 21 parameters were analyzed. The teleradiographs were made respecting the standard conditions provided for the execution of a correct radiogram: maximum intercuspation between the arches, Frankfurt plane parallel to the floor, lips in light contact, distance between focus and subject of 154 cm. On each teleradiography, the cephalometric tracing was performed by two operators (D.J. and R.G.), using acetate sheets, a pencil with a 0.5 mm diameter tip, and a desktop diaphanoscope. The investigators underwent an intra-examiner reliability check. Correlations (Pearson) between measurements on these occasions were 0.995 (*p* < 0.0001). Structural and esthetic cephalometric variables are reported in Table 1 and Table 2.

### 2.3. Statistical Analysis

The cephalometric measurements before and after therapy of 65 patients were collected in a table in the Microsoft Excel program for statistical analysis. Data analysis was performed using IBM’s SPSS software (version 25.0) (Statistical Package for Social Science), using three types of tests. The *t*-test on dependent samples was used to make comparisons of measurements within each group at the beginning and at the end of therapy. The single sample *t*-test made it possible to compare the results of each device, considered separately, with the control group, in order to detect statistically significant data. Using the univariate variance test with Fisher F statistic we compared the results of the three devices to each other to highlight any statistical significance.

## 3. Results

The data concerning the significance of the differences in the variables considered between the time T2 and T1 are summarized in Table 3, Table 4 and Table 5, respectively, for the functional equipment FR-2, TB, and O-o-G^®^. The mean values with the corresponding standard deviations of the parameters considered in this study are shown, at the beginning (T1) and at the end of the treatment (T2) (*t*-test on dependent samples). Table 6 shows the average values with standard deviation (ds), at the time T1 and T2, of the control group. Table 7 shows the mean values of the differences (T2–T1) of the variables, with the respective ds, for the four groups of subjects analyzed (FR-2 group, TB group, O-o-G^®^group, and control group), and the level of significance given by the comparison of the results obtained between the control group and each single device (*t*-test for single sample).

### 3.1. T-Test for Single Sample

#### 3.1.1. Structural Variables

All the three devices obtained significant results (*p* < 0.001) for the reduction of the ANB angle and the Pal P/U1 angle. The data obtained show a comparable level of significance for the three devices in the reduction of the SNA angle, while only for the TB group and the FR-2 group, a significant average increase of the SNB angle was found (*p* < 0.001). The results indicate an increase in the IMPA angle with a significant *p* value in statistical terms only for the O-o-G^®^ group; however, the noticeable increase in the Art-Pg and Art-B values in the three groups, in relation to the control group, indicate significant mandibular growth as a structural effect of all the equipment. However, significant results with *p* < 0.001 were obtained only from the FR-2 group and the TB group, with an improvement of 2.64 and 2.68 mm, respectively, for the Art-Pg variables and 2.8 and 2.97 mm for Art-B. The O-o-G^®^ group achieved results with lower levels of significance and an improvement of 2.03 mm (Art-Pg) and 2.01 mm (Art-B). A notable reduction of both overjet and overbite was detected in all three groups of treated patients.

#### 3.1.2. Esthetic Variables

All the devices analyzed in the present study determined a significant retraction of the upper lip point with respect to the Ricketts esthetic line; the greatest effect was achieved by the O-o-G^®^ group (2.1 mm), followed by the TB group (1.71 mm). Both in the FR-2 group and in the O-o-G^®^ group, a retraction of the inferior labial point was also observed with respect to the line E, as occurs in the control group. On the contrary, the TB group showed an increase in this value (0.83 mm). Therapy with TB gave the most significant results (*p* < 0.001) for the Z angle and for the angle of the convexity. Only in patients treated with FR-2, we found a statistically significant increase in the thickness of the upper lip. Finally, we found a significant mean reduction in the thickness of the lower lip in the three groups of treated subjects, while in the control group, this value underwent an increase; the greatest net effect was in the TB group.

### 3.2. Analysis of Fisher

This analysis made it possible to verify the presence of any significance resulting from the comparison between the average values of the variables of the three devices before and after the treatment. Of the 21 parameters analyzed, only the variation in the angle of convexity, overjet, and overbite yielded greater results for the TB group compared to the other devices. In Table 8, the results obtained from the comparison of the three devices are summarized, reporting only the data that showed statistical significance (analysis of the univariate variance, with Fisher F statistic).

## 4. Discussion

The present study showed a clinically significant increase in the mandibular length, evaluated by measuring the distances Art-Pg and Art-B, after therapy with all three devices, although TB and FR-2 achieved more significant results and a greater net effect than Occlus-o-Guide^®^. Other studies have shown similar results in increasing the Art-Pg distance after therapy with TB [17,18,19,20,21]. A meta-analysis on the skeletal effects at the jaw level in patients undergoing treatment with FR-2, conducted by Perillo and others, found a statistically significant outcome on the mandibular growth produced by this device [22]. The increase in mandibular length is probably responsible for the increase in the SNB angle obtained in the three groups of treated patients but reached statistical significance levels only in the TB and FR-2 group. Similar results are reported in the literature by Mills for TB and by Perillo and others for FR-2 [17,23]. In fact, not all studies, such as that conducted by De Almeida et al., have observed a significant increase in SNB in patients treated with FR-2 [24]. However, systematic review and meta-analysis showed that treatment effects of removable functional appliances in patients with Class II malocclusion are largely dentoalveolar rather than skeletal [25].

The present study evaluated the impact of the three devices on maxillary growth by increasing the distance Art-A and the possible variation of the SNA angle. The reduction of the SNA angle was minimal for the three groups. The results, however, agree with the literature showing a unanimous opinion about the low clinical relevance of the reduction of SNA following Class II functional therapy. The increase in Art-A distance for both the TB and O-o-G^®^ groups is very similar to that found in the control group. We cannot exclude that the results obtained indicate a restriction, albeit minimal, in the maxillary development produced by the two devices, whereas greater growth of the upper jaw occurs in Class II malocclusions [26]. An important indicator of the maxilla/jaw ratio is the value of the ANB angle that underwent on average a significant reduction of about 2° in all groups of subjects treated. Several authors, including Šidlauskas, have observed comparable results for TB [18]; others, such as Perillo et al. have observed comparable results for FR-2, and similar data emerged from clinical cases resolved with O-o-G^®^ from Laganà and Cozza [23,27]. However, from a study conducted through a systematic review of the literature, Koretsi and others judge the minimal and often negligible clinical relevance of skeletal effects [25]. Results from the present study have detected a palatal tipping of the upper incisors greater than the vestibularization of the lower incisors, whose effect plays a fundamental role in reducing overjet, as claimed by numerous authors, including Janson et al. [28,29]. Mills states that improvement in mandibular development may not be fully evaluated if growth is expressed in a predominantly vertical direction [30]. From the analysis performed, we found an increase in the anterior facial height, especially in patients treated with TB and with FR-2, but the proportion between the lower and total anterior facial height was not affected by the treatment. The results obtained show the ability of all three devices to achieve a significant reduction in overjet and overbite, as provided by literature data [29,31].

Studies that evaluate the esthetic results are limited for many devices, as for the FR-2 and the O-o-G^®^. Patients with Class II malocclusion usually have a convex facial profile and a retraction of the soft tissue pogonion. A very significant decrease in the convexity of the facial profile emerged in the group of subjects treated with TB, evaluated thanks to the increase in the angle of the convexity (CON angle) and very similar results were found by Baysal and Uysal and other authors [6,32,33]. A significant increase in the angle Z was also observed in the three groups of subjects treated, especially in the TB group, a direct consequence of the anterior displacement of the soft tissue pogonion. Similar values were also observed by Varlik et al. [34]. Significant retrusion of the upper lip from the control group was observed in treated patients, particularly for TB and O-o-G^®^. This result is consistent with most of the data reported in the literature available for TB, as also argued by Quintão and others, although Morris et al. did not show a significant difference in the position of the upper lip in patients treated with such a device [35,36,37,38]. Roos has identified a relevant degree of correlation between the variation of the upper labial point distance E and the palatal tipping of the upper incisors [39]. In the present study, we observed a tendency toward palatal tipping of the upper incisors, more relevant in the TB group than in the others; instead, the retrusion of the upper lip achieved a greater net effect in the O-o-g^®^ group; probably other characteristics of the device are decisive in determining a retrusion of the upper lip. The moving forward of the lower lip resulted in statistical significance only for the TB group, in accordance with the results of Quintão et al. although this result is not referable to dental effects [35]. Other authors, on the other hand, did not identify a variation of this statistically relevant parameter in patients treated with TB [6,34]. The significant reduction in the thickness of the lower lip after TB therapy is in agreement with the claims of Baysalet al. [6]. Mandibular advancement, especially with TB, facilitates the correct distribution of the perioral soft tissues and guarantees correct labial contact and, therefore, a reduction in the thickness of the lower lip, which is often excessively altered in Class II malocclusion. The results obtained in the study show that the TB is the device among those considered that guarantees the best skeletal, dental (in the reduction of overjet), and esthetic results.

## 5. Conclusions

The structural analysis shows that the following:All three devices resulted in a significant increase in mandibular length, with higher results for FR-2 and TB.The reduction of the ANB angle was similar in the three groups, but the increase in the SNB angle was significant only for FR-2 and TB.The same levels of significance for the three devices were highlighted in the reduction of overjet and overbite in relation to the control group, but the reduction produced by TB was significant compared to that for the other two devices.The IMPA angle increased more in the O-o-G^®^ group.

The esthetic analysis shows the following:The esthetic evaluation showed overall more significant results in the TB group, especially in relation to the reduction of facial convexity.The retrusion of the upper lip was on average more significant in the O-o-G^®^ group, followed by that in the TB group.The thickness of the lower lip underwent the most significant reduction in patients treated with TB, but only FR-2 resulted in a greater increase in the thickness of the upper lip compared to the control group.A limitation of the study is represented by the fact that Occlus-o-Guide^®^ group has a lower mean age than the other two groups, so the skeletal variables at time T1 are not homogeneous between the three groups.

## Figures and Tables

**Table 1 ijerph-18-06978-t001:** Structural cephalometric variables.

SNA	Angle between SN and the N-A line, indicates the position of the maxilla on the sagittal plane
SNB	Angle between SN and the N-B line, indicates the position of the mandible on the sagittal plane
ANB	Difference between SNA an SNB, indicates skeletal class
SN^GoGn	Angle of the total divergence, determined by the intersection of the SN plane with the GoGn plane
max^man	Indicates the divergence of the mandible with respect to the maxilla
IMPA	Angle between the most protruding lower incisor axis and the mandibular plane
Pal P/U1	Angle between the axis of the most prominent upper incisor and the palatal plane
Art-B	Distance between the points Art and B, indicates the mandibular length
Art-Pg	Distance between the points Art and Pg, indicates the mandibular length
Art-A	Distance between the points Art and A, indicates the length of the upper jaw
N-Me	Distance between the points N and Me, indicates total frontal facial height
ANS-Me	Distance between the front nasal spine and the point Me, indicates the lower frontal facial height
OVJ	Indicates in the sagittal plane the position of the upper incisors with respect to the inferior ones
OVB	Indicates in the vertical plane the position of the upper incisors with respect to the lower ones

**Table 2 ijerph-18-06978-t002:** Esthetic cephalometric variables.

Z	Angle between the Frankfurt floor and the Z line
CON	Angle formed between soft tissue nasion, subnasale, and soft tissue pogonion
ULE	Distance from the labrale superioris to a line joining the nasal tip and the soft tissue pogonion
LLE	Distance from the labrale inferioris to a line joining the nasal tip and the soft tissue pogonion
LLT	Distance between the vermilion point and the labial surface of the maxillary incisor
ULT	Distance between the labrale inferioris and the most prominent buccal point of the lower incisors
ULL	Vertical distance between the upper lip stomion and the subnasale

**Table 3 ijerph-18-06978-t003:** Data collection of the differences in the variables considered between the time T1 and T2 for Frankel.

	Frankel T2–T1			
	Mean	DS	Significance	
SNA	−0.09	0.996	0.68	NS
SNB	2.13	0.97	*p* < 0.001	***
ANB	−2.17	0.78	*p* < 0.001	***
SN/Man	−0.57	2.86	0.28	NS
Max/Man	−0.48	1.83	0.208	NS
IMPA	2.09	3.26	0.006	**
Pal P/U1	−3.3	3.94	0.001	**
Art-A	3.3	2.06	*p* < 0.001	***
Art-Pg	6.74	2.68	*p* < 0.001	***
Art-B	6	3.12	*p* < 0.001	***
N-Me	7.13	4.16	*p* < 0.001	***
ANS-Me	4.39	2.89	*p* < 0.001	***
OVJ	−3.17	1.92	*p* < 0.001	***
OVB	−1.65	1.87	0.001	**
ULE	−1.39	2.02	0.004	**
LLE	−0.35	1.67	0.404	NS
Z	3.22	5.1	0.006	**
CON	1.17	3.59	0.108	NS
LLT	−0.13	1.74	0.814	NS
ULT	2	1.3	*p* < 0.001	***
ULL	1.28	2.28	0.013	*

* *p* < 0.05; ** *p* < 0.01; *** *p* < 0.001.

**Table 4 ijerph-18-06978-t004:** Data collection of the differences in the variables considered between the time T1 and T2 for Twin Block.

	Twin Block T2–T1			
	Mean	DS	Significance	
SNA	−0.17	0.86	0.298	NS
SNB	2.28	1.07	*p* < 0.001	***
ANB	−2.33	1.03	*p* < 0.001	***
SN/Man	0.33	2.03	0.428	NS
Max/Man	0.2	1.2	0.08	NS
IMPA	2.39	3.6	0.012	*
Pal P/U1	−5.44	2.48	*p* < 0.001	**
Art-A	2.61	0.92	*p* < 0.001	***
Art-Pg	6.78	1.55	*p* < 0.001	***
Art-B	6.17	0.98	*p* < 0.001	***
N-Me	5.22	1.73	*p* < 0.001	***
ANS-Me	3.78	1.44	*p* < 0.001	***
OVJ	−4.28	0.89	*p* < 0.001	***
OVB	−2.22	1.06	*p* < 0.001	***
ULE	−1.94	1.16	*p* < 0.001	***
LLE	0.5	0.98	0.13	NS
Z	5.17	3.29	*p* < 0.001	***
CON	4.06	2.31	*p* < 0.001	***
LLT	−1.06	1.66	0.032	*
ULT	1.33	1.64	0.003	**
ULL	0.42	1.88	0.377	NS

* *p* < 0.05; ** *p* < 0.01; *** *p* < 0.001.

**Table 5 ijerph-18-06978-t005:** Data collection of the differences in the variables considered between the time T1 and T2 for Occlus-o-Guide^®^.

	Occlus-o-Guide^®^ T2–T1			
	Mean	DS	Significance	
SNA	−0.58	1.89	0.178	NS
SNB	1.42	2.08	0.002	**
ANB	−1.96	1.12	*p* < 0.001	***
SN/Man	−0.38	2.81	0.531	NS
Max/Man	−1.42	3.28	0.038	*
IMPA	3.42	3.97	*p* < 0.001	***
Pal P/U1	−4.17	6.58	0.006	**
Art-A	2.42	1.32	*p* < 0.001	***
Art-Pg	6.13	3.11	*p* < 0.001	***
Art-B	5.21	2.93	*p* < 0.001	***
N-Me	6.5	3.88	*p* < 0.001	***
ANS-Me	3.17	3.14	*p* < 0.001	***
OVJ	−3.13	1.85	*p* < 0.001	***
OVB	−1.04	1.27	0.002	**
ULE	−2.33	2.2	*p* < 0.001	***
LLE	−0.17	2.46	0.447	NS
Z	3.17	5.12	0.002	**
CON	1.21	2.75	0.037	*
LLT	−0.58	1.66	0.273	NS
ULT	0.88	1.77	0.054	NS
ULL	1	1.56	0.002	**

* *p* < 0.05; ** *p* < 0.01; *** *p* < 0.001.

**Table 6 ijerph-18-06978-t006:** Data collection of the differences in the variables considered between the time T1 and T2 for the control group.

	Control Group T2–T1	
	Mean	DS
SNA	0.41	0.4
SNB	0.7	0.5
ANB	−0.2	0.1
SN/Man	−0.7	0.6
Max/Man	−0.6	0.4
IMPA	0.7	0.5
Pal P/U1	2.4	1.2
Art-A	2.4	1.7
Art-Pg	4.1	3.2
Art-B	3.2	1.9
N-Me	3.7	1.8
ANS-Me	2.1	0.9
OVJ	0.1	0.1
OVB	0.3	0.6
ULE	−0.23	1.36
LLE	−0.33	2.26
Z	−0.14	3.86
CON	0.12	2.67
LLT	0.73	2.3
ULT	1.33	0.15
ULL	0.42	1.59

**Table 7 ijerph-18-06978-t007:** Comparison of mean differences between treated and control subjects.

	Frankel		Twin Block		Occlus-o-Guide		Control Group		Frankel/		Twin Block/		Occlus-o-Guide/	
	T2–T1		T2–T1		T1		T2–T1		Control Group		Control Group		Control Group	
	Mean	DS	Mean	DS	Mean	DS	Mean	DS	Significance		Significance		Significance	
SNA	−0.09	0.996	−0.17	0.86	−0.58	1.89	0.41	0.4	0.026	*	0.011	*	0.017	*
SNB	2.13	0.97	2.28	1.07	1.42	2.08	0.7	0.5	*p* < 0.001	***	*p* < 0.001	***	0.105	NS
ANB	−2.17	0.78	−2.33	1.03	−1.96	1.12	−0.2	0.1	*p* < 0.001	***	*p* < 0.001	***	*p* < 0.001	***
SN/Man	−0.57	2.86	0.33	2.03	−0.38	2.81	−0.7	0.6	0.823	NS	0.045	*	0.576	NS
Max/Man	−0.48	1.83	0.2	1.2	−1.42	3.28	−0.6	0.4	0.753	NS	0.04	*	0.235	NS
IMPA	2.09	3.26	2.39	3.6	3.42	3.97	0.7	0.5	0.054	NS	0.06	NS	0.003	**
Pal P/U1	−3.3	3.94	−5.44	2.48	−4.17	6.58	2.4	1.2	*p* < 0.001	***	*p* < 0.001	***	*p* < 0.001	***
Art-A	3.3	2.06	2.61	0.92	2.42	1.32	2.4	1.7	0.031	*	0.324	NS	0.951	NS
Art-Pg	6.74	2.68	6.78	1.55	6.13	3.11	4.1	3.2	*p* < 0.001	***	*p* < 0.001	***	0.004	**
Art-B	6	3.12	6.17	0.98	5.21	2.93	3.2	1.9	*p* < 0.001	***	*p* < 0.001	***	0.003	**
N-Me	7.13	4.16	5.22	1.73	6.5	3.88	3.7	1.8	0.001	**	0.002	*	0.002	**
ANS-Me	4.39	2.89	3.78	1.44	3.17	3.14	2.1	0.9	0.001	**	*p* < 0.001	***	0.11	NS
OVJ	−3.17	1.92	−4.28	0.89	−3.13	1.85	0.1	0.1	*p* < 0.001	***	*p* < 0.001	***	*p* < 0.001	***
OVB	−1.65	1.87	−2.22	1.06	−1.04	1.27	0.3	0.6	*p* < 0.001	***	*p* < 0.001	***	*p* < 0.001	***
ULE	−1.39	2.02	−1.94	1.16	−2.33	2.2	−0.23	1.36	0.01	*	*p* < 0.001	***	*p* < 0.001	***
LLE	−0.35	1.67	0.5	0.98	−0.17	2.46	−0.33	2.26	0.96	NS	0.002	**	0.748	NS
Z	3.22	5.1	5.17	3.29	3.17	5.12	−0.14	3.86	0.005	**	*p* < 0.001	***	0.001	**
CON	1.17	3.59	4.06	2.31	1.21	2.75	0.12	2.67	0.173	NS	*p* < 0.001	***	0.065	NS
LLT	−0.13	1.74	−1.06	1.66	−0.58	1.66	0.73	2.3	0.027	*	*p* < 0.001	***	0.001	**
ULT	2	1.3	1.33	1.64	0.88	1.77	1.33	0.15	0.023	*	0.993	NS	0.222	NS
ULL	1.28	2.28	0.42	1.88	1	1.56	0.42	1.59	0.084	NS	0.994	NS	0.082	NS

* *p* < 0.05; ** *p* < 0.01; *** *p* < 0.001.

**Table 8 ijerph-18-06978-t008:** Significant values between three treated groups.

	Frankel		Twin Block		Occlus-o-Guide		Twin Block/		Twin Block/	
	T2–T1		T2–T1		T2–T1		Frankel		Occlus-o-Guide	
	Mean	DS	Mean	DS	Mean	DS	Significance		Significance	
OVJ	−3.17	1.92	−4.28	0.89	−3.13	1.85	0.024	*	0.02	*
OVB	−1.65	1.87	−2.22	1.06	−1.04	1.27	0.058	NS	0.041	*
CON	1.17	3.59	4.06	2.31	1.21	2.75	0.004	**	0.009	**

* *p* < 0.05; ** *p* < 0.01.

## Data Availability

The data underlying this article are available in the article.

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
