# Peer review of "Comparative Evaluation of Esthetic and Structural Aspects in Class II Functional Therapy. A Case–Control Retrospective Study"

_ijerph, 2021, doi:10.3390/ijerph18136978_

Round 1
Reviewer 1 Report
The article traces a interesting comparative evaluation of aesthetic and structural aspects in the class II functional therapy. The methods are properly conducted. The availability of data adheres to the expected standards of your research community. The claims are appropriately discussed in the context of previous literature, although this is outdated. The manuscript is clearly written. There are no special ethical concerns. Plagiarism detector program showed low content similarity (2.38%), guaranteeing the originality of this review article. Article with merit for publication and compatible proposal within the scope of the International Journal of Environmental Research and Public Health. Point-to-point instructions for improving the text follow in the comments to the author. After major corrections to the form and content of this version, the manuscript will be ready for publication.
Title:
- The title of the manuscript is succinct and clear.
Abstract:
- The abstract can be improved. The performance of dental changes among the therapies is not mentioned, as recorded in the beginning in “Background”.
- Material and Methods:
- Considering that the scope of the journal is focused on Public Health, authors should at least create a paragraph contextualizing the relevance of the therapeutic proposals. Explore the potential for application of these, considering operational costs, ease of handling by specialist and expected results.
- Material and Methods:
- The present study adopted different amounts of patients in the three groups proposed within the sample of 65 treated patients and 20 patients in the control group. Authors should better explain why in this difference. In case of drop-out, justify in the text the lack of clinical continuation of these patients.
- Results:
- Table 7 (line 129) is unconfigured, review the data presentation.
- Discussion:
- The claims are appropriately discussed in the context of previous literature, although it is very outdated. Support the results on the current literature.
- Conclusions:
- The conclusion topics deserve to be rewritten, as expected in this section. Aggregate the findings within the categories of function and aesthetics in view of the different therapeutic protocols, to facilitate the reader´s general understanding of the contribution of this research in the field of study of functional jaw orthopedics.
- Still, there were no cited limitations of the study or potential of future studies for the theme addressed, of interest to Public Health.
- Remove the meaningless text between lines 253-255:
“6. Patents:
This section is not mandatory but may be added if there are patents resulting from the work reported in this manuscript.
References:
- Many references are out of the journal's standard. Review this section completely.
- The article presents outdated references. Among the 30 references with the mentioned year, only 4 or 13% of this total was published in the last 5 years (2016-2021). Insert more recent references.
Reviewer 2 Report
This manuscript described comparative evaluation of esthetic and structural aspects in the Class II functional therapy. A case-retrospective study. The authors summarized that all three functional appliances have proven to be effective in resolving skeletal changes and improving facial esthetics.
Overall this review is an interesting area for clinical orthodontics. The authors compared the dental and skeletal changes after 3 functional appliance treatment in the patients with Class II div1 using 2D cephalometric analysis. The significant differences were found in certain parameters of the cephalometric analysis.
The questions remain to be clarified.
- How did the authors avoid the selection bias in the study?
- Could the authors demonstrate the percentage of contribution in the correction of overjet by skeletal change and dentoalveolar change among 3 appliances?
- Could the authors discuss why O-o-G appliance could not induce the increase of Art-B or Art-pog compared to the other two appliances.
Thank you for the opportunity to review this manuscript.
Round 2
Reviewer 1 Report
The article traces a interesting comparative evaluation of aesthetic and structural aspects in the class II functional therapy. The methods are properly conducted. The availability of data adheres to the expected standards of your research community. The claims are appropriately discussed in the context of previous literature. The manuscript is clearly written. There are no special ethical concerns. Plagiarism detector program showed low content similarity, guaranteeing the originality of this article. Article with merit for publication and compatible proposal within the scope of the International Journal of Environmental Research and Public Health. The manuscript in this second version is ready for publication.